# Effect of Heat Transfer Driving Force and Ice Seed Loading on the Production of Ice and Salt from a Dilute Brine Treated Using Eutectic Freeze Crystallization

Anotidaishe Spencer, Jemitias Chivavava and Alison Emslie Lewis *

Crystallization and Precipitation Research Unit, Chemical Engineering Department, University of Cape Town, Cape Town 7700, South Africa
* Correspondence: alison.lewis@uct.ac.za

**Abstract:** Eutectic Freeze Crystallization (EFC) is a separation technology that separates solute from solvent by cooling the brine to a temperature below its eutectic point, such that ice and salt simultaneously crystallize out of the solution. Achieving consistent production of ice and salt at high production rates has been a challenge for EFC. This is due to heat transfer limitations, which are more severe when EFC is applied to dilute brines. This work investigated the effect of the heat transfer driving force, $\Delta T_{LMTD}$, and ice seed loading (SL) on the production of ice and salt from a dilute brine. A 1.45 L stirred crystallizer was used for the experiments at varying coolant temperatures to investigate the effect of $\Delta T_{LMTD}$, and at varying seed masses to investigate the effect of seed loading. It was found that, as the $\Delta T_{LMTD}$ increased, the yield of ice and salt increased. This was attributed to the increase in the heat transfer rate as $\Delta T_{LMTD}$ and heat transfer rate are directly related. The ice yield was divided between ice in suspension and ice formed on the wall (scale layer), with a majority of the total ice yield being scale ice. Increasing the seed loading (SL) increased the yield of ice in suspension and decreased the yield of scale ice. The seeds allowed for increased surface area for crystallization in the bulk. This allowed for most of the supersaturation to be consumed in the bulk, leaving little supersaturation for crystallization at the wall. This reduced the propensity for scale formation. The reduction in the scale layer increased the heat transfer rate between the bulk and the coolant, allowing for more ice to be formed in suspension.

**Keywords:** Eutectic Freeze Crystallization; heat transfer; seeding; scale formation



## 1. Introduction

The worldwide increase in desalination and mine water treatment using reverse osmosis has led to an increase in the generation of hypersaline brine streams [1]. This is increasingly being recognized as a global issue and extensive research has been undertaken to address the problem [2]. Eutectic Freeze Crystallization (EFC) is a novel separation technology that can be used for treating hypersaline brine streams. It works by cooling the stream to the eutectic temperature, at which point both ice and salt (or acid) will crystallize. The ice, being less dense than the solution, floats, and the salt, being denser, will sink – thus effecting a gravity separation [3]. The selectivity of crystal growth leads to very pure products even when the solution contains many impurities, as is often the case for industrial streams [4]. EFC has several advantages over competing techniques. For example, the energy required to freeze water is six times less than that for vaporization [5], no additional chemicals are required, operation at lower temperatures is safer, and there is potential for zero-liquid discharge ZLD [3]. In EFC, both ice and salt crystallize out of solution, and therefore have the potential to be recovered as separate products. The hypersaline brine that is generated by a reverse osmosis process can be treated by EFC; EFC can also be combined with other separation technologies to optimize separation processes [6].

## 2. Background

Eutectic Freeze Crystallization has been implemented on real industrial brines [7] and performs well when the brines are highly concentrated. However, when applied to dilute brines, various challenges have been encountered, including excessive ice scaling, and difficulty in achieving consistent production. In order to overcome these challenges, a better understanding of how the operating conditions of the EFC crystallizer influence the ice scaling and the production rate is vital. The ability to control ice scaling and to achieve consistent production would allow for better application and control of the technology such that efficient, economical operation as well as consistently high rates of production can be achieved. This work focusses on the factors affecting ice scaling and production rate in particular.

Eutectic Freeze Crystallization is thermodynamically driven by supersaturation. For EFC, supersaturation is the deviation of the solution temperature from the equilibrium temperature at a given concentration. Supersaturation is provided to a solution by cooling; therefore, the production capacity of the EFC process depends on heat transfer rate from the solution to the coolant. When the heat transfer rate is compromised, the production rate becomes limited. Many authors have stressed that, in EFC, the most important design characteristics are the heat transfer rate, which dictates the production rate, and the separation efficiency [8–11]. This is depicted in the relationship between heat transfer rate and production rate in Equation (1).

$$\dot{m} = \frac{Q}{\Delta H_{crystallization}} \tag{1}$$

where $\dot{m}$ (kg s$^{-1}$) is the mass flow rate of the products i.e., production rate, $Q$ is the heat transfer rate (W = kJs$^{-1}$) and $\Delta H_{crystallization}$ (kJ kg$^{-1}$) is the heat of crystallization of both ice and salt. For a batch process, these quantities would be the totals over the whole batch time.

The heat transfer rate is a function of the heat transfer area, $A$ (m$^2$), the heat transfer coefficient, $U$ (W m$^{-2}$ K$^{-1}$), and the heat transfer driving force, $\Delta T_{LMTD}$ (K) as shown in Equation (2).

$$Q = UA\Delta T_{LMTD} \tag{2}$$

Therefore, the heat transfer rate can be increased by increasing the heat transfer area, $A$, which is a function of the crystallizer design [10,12]. It can also be improved by increasing the heat transfer coefficient, $U$, which is a function of the flow characteristics and physical properties of both the coolant and of the process fluid/slurry, as well as thermal properties of the material of construction.

The overall heat transfer coefficient ($U$) is a function of the heat transfer coefficient on the coolant side, $\alpha_{coolant}$ (W m$^{-2}$ K$^{-1}$), process side, $\alpha_{process}$ (W m$^{-2}$ K$^{-1}$) and that of the wall, $\alpha_{wall}$ (W m$^{-2}$ K$^{-1}$). The overall thermal resistance $\frac{1}{U}$ can be calculated using Equation (3).

$$\frac{1}{U} = \frac{1}{\alpha_{process}} + \frac{1}{\alpha_{coolant}} + \frac{1}{\alpha_{wall}} \tag{3}$$

where $\alpha$ can be described by Equation (4):

$$\alpha = \frac{1}{RA} \tag{4}$$

where $R$ (K/W) is the thermal resistance to heat transfer, which is determined by the type of heat transfer (conductive or convective), material properties (thermal conductivity) and heat transfer configuration (rectangular, cylindrical, or spherical), and $A$ (m$^2$) is the heat transfer area.

If a scale layer is formed in the crystallizer, then Equation (3) can be modified to include the effect of the scale layer on the overall heat transfer resistance, as depicted in Equation (5).

$$\frac{1}{U} = \frac{1}{\alpha_{process}} + \frac{1}{\alpha_{coolant}} + \frac{1}{\alpha_{wall}} + \frac{1}{\alpha_{scale}}$$ (5)

where $\alpha_{scale}$ is the heat transfer coefficient of the scale (W m$^{-2}$ K$^{-1}$).

The heat released by crystallization is transferred to the coolant. The heat transferred to the coolant can be calculated using Equation (6)

$$Q_{cool} = \dot{m}_{coolant} Cp_{coolant} (T_{coolant,in} - T_{coolant,out})$$ (6)

where $\dot{m}_{coolant}$ (kg s$^{-1}$) is the mass flow rate of the coolant, $Cp$ (J kg$^{-1}$K$^{-1}$) is the heat capacity of the coolant and $T$ (K) is the temperature of the coolant.

In cases where the thermal resistance of the coolant is rate-limiting, higher coolant flow rates result in higher values of U and, in turn, higher heat transfer rates [13]. Since U, A and $\Delta T_{LMTD}$ affect heat transfer, they also indirectly affect the production rate of ice and salt. Studies on factors affecting U and A have been carried out [4,13,14].

The effect of $\Delta T_{LMTD}$ on the production rate in EFC has been studied [12,13]. However, these studies have been for very concentrated brines. The effect of $\Delta T_{LMTD}$ on production rate of ice and salt from dilute brines has not been studied. This was investigated in this study to fill this gap since it is not always possible to preconcentrate a dilute brine before treatment.

The investigation was carried out via a seeded Eutectic Freeze Crystallization process. Seeding is the addition of solute or solvent crystals to a supersaturated solution such that primary nucleation is avoided and either growth on the surface of ice crystals or secondary nucleation is favored. Seeds provide surface area for the consumption of supersaturation and thereby avoid the high supersaturations that would lead to primary nucleation [15]. In practice, in order to avoid the unwanted consequences of a primary (heterogeneous) nucleation event, seeds are added at the beginning of the batch process, within the metastable zone. The supersaturation profile of the process depends on the amount and size of the seed particles, as well as on the rate of cooling. If appropriate seed addition and rate of cooling are applied, the solution never reaches the metastable limit and primary nucleation does not occur throughout the batch process [15].

The mass of seeds added to a crystallizing system is known as the seed loading. A measure of the seed loading is the seed loading ratio. The seed loading ratio, $C_s$, is the ratio of the mass of seeds to the theoretical yield of the salt being crystallized [16]. This is depicted in Equation (7) below:

$$C_S = \frac{W_S}{W_{th}}$$ (7)

where $W_s$ is the mass of the seeds (kg) and $W_{th}$ is the theoretical yield (kg).

For every species, there exists a seed loading ratio where the seed mass allows for growth, such that nucleation in the bulk is avoided and from which a unimodal product distribution can be generated [16]. This is known as the critical seed loading ratio, $C_s^*$. Doki and co-workers [17] developed an equation relating $C_s^*$ and $L_s$, the initial seed mean size (μm). This is shown in Equation (8).

$$C_s^* = 2.17 \times 10^{-6} L_s^2$$ (8)

Crystallization carried out using $C_S > C_S^*$ results in growth of the seeds, whilst using $C_S < C_S^*$ results in both growth of the seeds and nucleation in the bulk/wall [17]. The critical surface area is necessary to promote growth and avoid nucleation in the bulk and can be calculated using a combination of initial seed mean size ($L_s$) and seed mass at or above the critical seed mass, $W_s^*$. The equations above were developed for salt seeding. This work focuses on ice seeding, which is also of interest because although the ice seeds provide surface area for consumption of supersaturation, they are not the only source of

surface area in the crystallizer. The cooled wall also provides surface area for crystallization and/or adhesion. Crystallization or adhesion onto the wall results in an ice scale layer which limits the efficiency of heat transfer from the bulk to the coolant. This reduces the yield of ice and salt in EFC. Eventually, the scale layer can become too thick for the scrapers to remove [13]. This means that downtime is required for removal of the scale layer, which results in low productivity. Seeds may allow for the control of the site of supersaturation consumption (wall or bulk), such that high production rates and production consistency is improved. Providing the system with enough surface area in the bulk, via the ice seed loading, favors preferential crystallization in the bulk rather than on the wall. This concept of decrease in ice scaling on the wall due to the availability of ice in the bulk, which promotes crystallization in suspension by providing surface area for growth and by inducing secondary nucleation, has been developed by Leyland and collaborators [18] based on observation but has not been quantified. This study was aimed at quantifying the relationship between ice seed loading and ice crystallization in the bulk and on the wall.

In this work, the effect of $\Delta T_{LMTD}$ and ice seed loading on the yield of products and specifically, the partitioning of the ice product between the wall (scale) and the bulk (suspension) from a dilute brine was investigated and quantified.

## 3. Materials and Methods

### 3.1. Experimental Design

A binary $Na_2SO_4$ solution was made from 4 wt.% chemical grade $Na_2SO_4$; 96 wt.% deionized water was prepared and used for the experiments. 4 wt.% is the eutectic concentration of a binary $Na_2SO_4$ solution as determined by OLI Stream Analyser[TM] 10.0 [19] software. This allowed for the study to be carried out under eutectic conditions. Though a 4 wt.% $Na_2SO_4$ aqueous solution is almost at eutectic concentration, it is nonetheless a dilute solution in terms of TDS. When a brine contains a salt at its eutectic concentration and temperature, such a brine cannot be further concentrated before EFC can be applied. This allowed for the findings from the study to fill the gap in literature on the application of EFC on dilute brines.

The effect of $\Delta T_{LMTD}$ on the yield of ice and on the split of the ice product between the bulk and wall was investigated for a range of $\Delta T_{LMTD}$ values between 2 and 10 °C (2, 4, 6, 8 and 10 °C). This range was based on preliminary experiments that showed that above $\Delta T_{LMTD} = 10$ °C excessive ice scaling occurred such that the crystallizer could not be operated. For the $\Delta T_{LMTD}$ experiments, a constant seed loading of 6 wt.% (Ws/v) was used as this was found to be a seed loading which produced large crystals [20]. Large crystals are ideal for improved separation of ice and salt.

In the second set of experiments, the effect of seed loading on the yield of ice on the wall (ice scale) and in suspension was investigated. In order to select the range of ice seed loadings, the critical seed loading ratio was calculated using Equation (8) and found to be 0.58. Equation (7), as well as theoretical yields obtained from OLI Stream Analyser[TM] 10.0 [20] software, was used to translate the critical seed loading ratio to critical seed loadings. These were found to range from 708 g–799 g, which translated to between 49 and 55% of the crystallizer volume. This was impractical to implement so ice seed loadings between 0 and 12 wt.% (0.1, 3, 6, 9 and 12 wt.%) were selected for experimental investigations as these were more practical and ranged equally below and above the 6 wt.% from the study by Kaboyashi and Shirai [20]. For these experiments, a $\Delta T_{LMTD}$ of 4 °C was used.

The seeds used in the experiments had to be of reproducible shape and size distribution to allow for observed changes in yield to be only caused by the change in seed loading. Since the surface area of ice seeds was varied by increasing the mass of the seeds while maintaining the CSD, the influence of ice seed loading on hydrodynamic conditions in the crystallizer and process side heat transfer coefficient was evaluated. Both the Reynolds number (Re) and the process side heat transfer coefficient, $\alpha_{process}$, were therefore estimated at the different seed loadings (see Appendix A for Re calculations). The Re was

also calculated at the end of each experiment to evaluate if there was further change in hydrodynamic conditions as the magma density increased during crystallization.

The investigations were conducted in batch mode because it was a simple system for investigating ice-seeding and could give informative results without the extensive challenges involved in preparing and handling ice seeds for a continuous system. For the purposes of this study, the batch mode was considered to be sufficiently informative to justify its choice. Operating in batch limited the evaluation of both Re and $\alpha_{process}$ as these could only be evaluated at the start and at the end of the experiment.

### 3.2. Experimental Set Up

Figure 1 shows a schematic of the setup of the equipment used for the investigation.

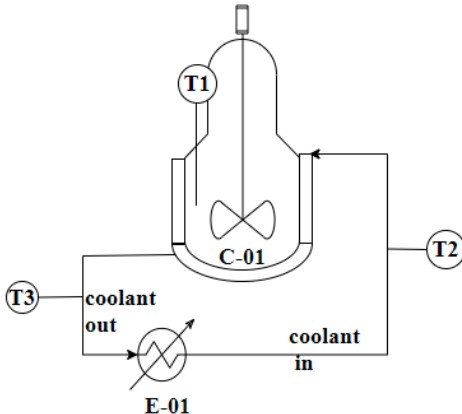

**Figure 1.** Schematic of batch crystallizer setup.

A 1.45 L glass, jacketed, stirred crystallizer, C-01, of diameter 0.11 m, was used with a Lauda Proline cooling unit, E 01, to cool the solution. Polydimethylphenylsiloxane (Kryo-51) was used as the coolant. Three platinum resistance temperature probes, connected to an F252 AC bridge and a 16-channel switchbox, were used to measure the temperatures of the solution (T1) and the coolant (T2 and T3). The temperatures were recorded by Ulog temperature logging software (V6.1). All experiments were carried out at room temperature, where room temperature was 21.15 °C ± 0.2, so the vessel and piping were insulated. Preliminary experiments showed that a pitched blade impeller with a diameter of 0.075 m ensured efficient suspension of the seeds. The impeller was positioned a third of the total crystallizer length above the base of the crystallizer to avoid dead zones

### 3.3. Experimental Procedure

It was desired that the size and shape of the seeds produced by the ice shaving machine be reproducible. To test for the reproducibility of the ice seed size and shape, three repeat seed preparation experiments were carried out before the seeds could be used in the main batch experiments. The seeds that were used in the experiments were prepared by freezing distilled water in ice cube trays overnight in a freezer at −18 °C. The ice cubes were emptied into the shaving chamber of an ice shaving machine. Once switched on, the ice cubes were shaved to fine ice crystals, and these were collected in a pre-cooled, insulated beaker. After collecting the seeds in the insulated beaker, a sample of ice seeds from many random positions in the beaker was collected and set under a light microscope for imaging. The slides used under the microscope were insulated and frozen at −18 °C prior to use under the microscope to minimise melting. The microscope was connected to a camera, which captured images using analySIS docu™ software. The images were saved for sizing and shape analysis. The size and shape characterizations were carried out using the captured images and ImageJ [21] image analysis software. ImageJ was used to obtain the area of each crystal and this area was converted from pixels to microns using a measurement of the scale bar from the images. The area was then converted to an equivalent diameter using

the formula for the area of a circle as the crystals were disc shaped. It was only possible to analyze up to 100 crystals for ice, as the imaging was limited by melting of the sample.

Figure 2 shows the crystal size distribution of the ice seeds. The d50 was calculated to be 520 μm. This is typically larger than ice crystals obtained in EFC, which are usually up to 250 μm under continuous conditions [4,12]. It can also be seen that the seed size distribution is reproducible due to the small error bars. Reproducibility of the shape was also a desired characteristic for the seeds. Figure 3 shows examples of the crystals obtained in the three repeat experiments. The images show that the crystals were typically disc shaped, which is expected for ice crystals [4].

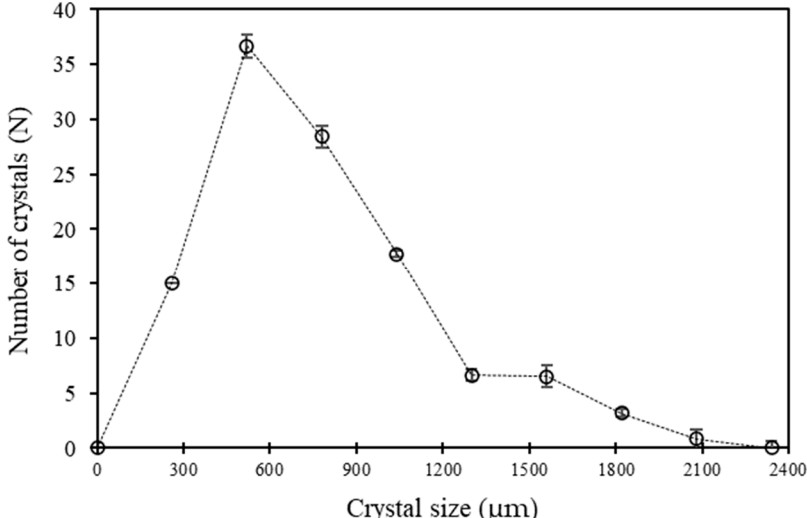

**Figure 2.** Crystal size distribution of the ice seeds formed from the ice shaving machine.

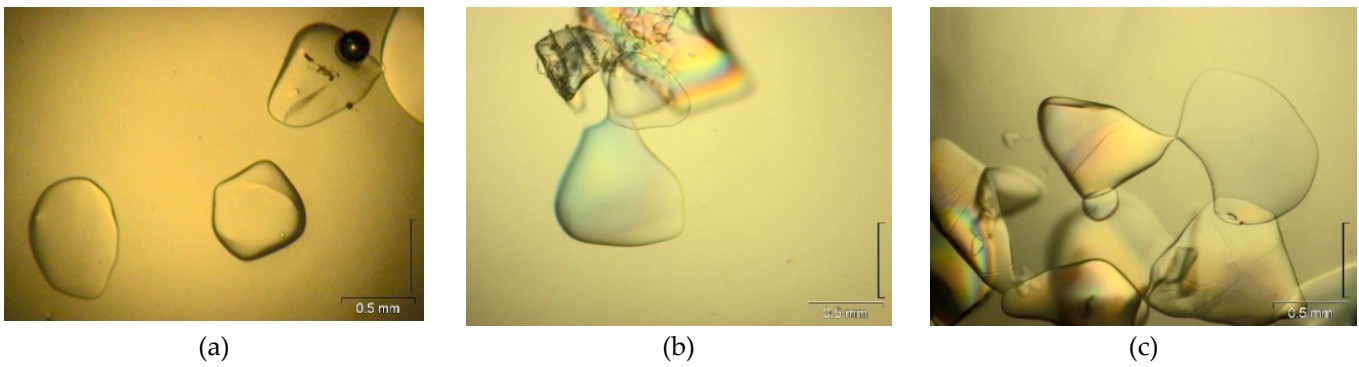

     (a)                   (b)                   (c)

**Figure 3.** Images of ice seeds from the three repeat experiments (**a–c**) for seed preparation using the ice-shaving machine.

A limitation in the size and shape analysis was in the imaging and was a function of ice movement during imaging. Figure 3 shows that the ice crystals were not always positioned in the same way under the microscope. In Figure 3a, the ice crystals are less attached to each other than in Figure 3b,c. The attachment of ice crystals while on the slides under the microscope was a function of ice movement. Ice movement was unavoidable as it was due to melting. The insulation and freezing of the slides reduced the melting but did not eradicate it entirely.

After these first three repeats, this seed preparation procedure was carried out prior to the batch crystallization without the size analysis.

The experimental procedure is illustrated in Figure 4 and described thereafter.

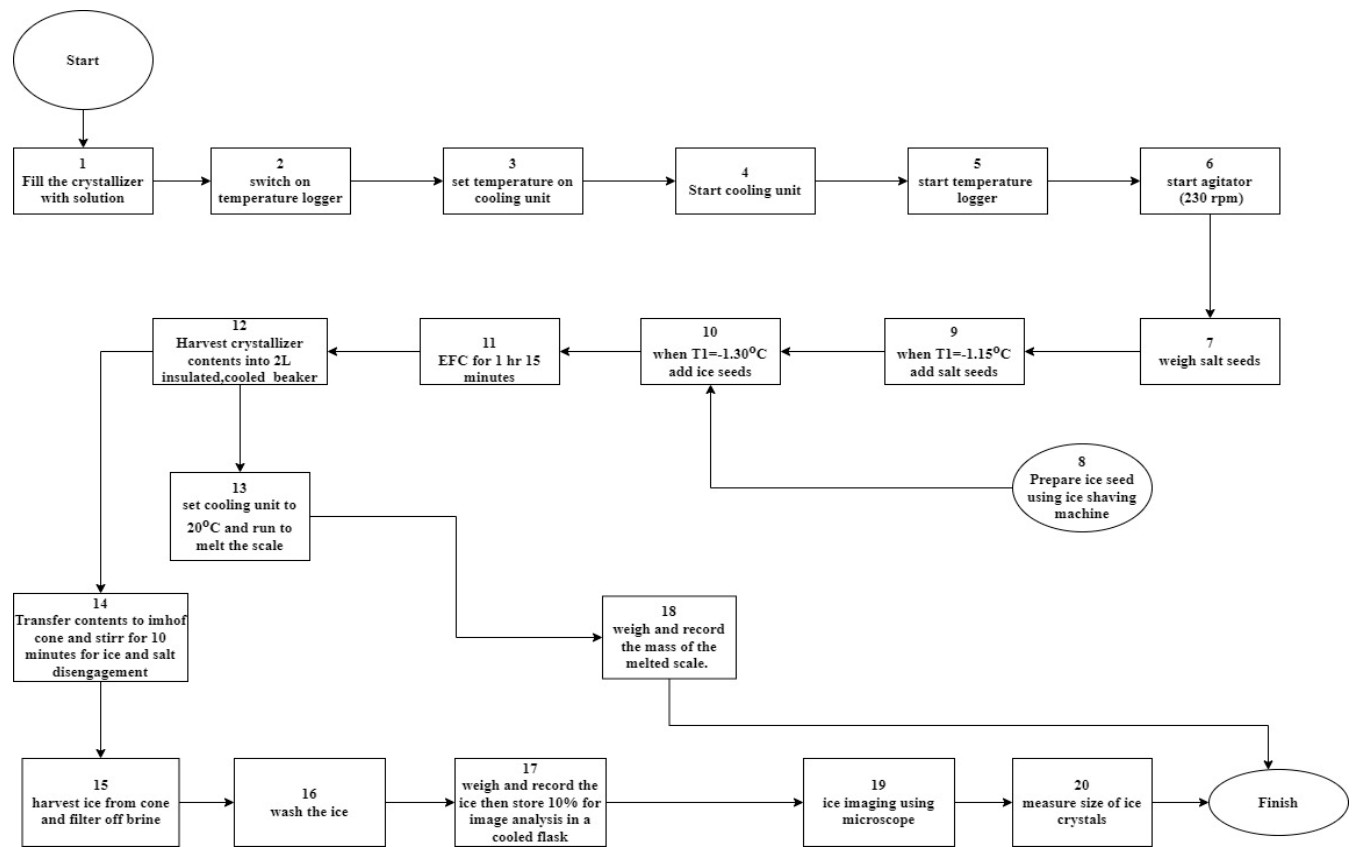

**Figure 4.** Experimental procedure flow chart.

To start each experiment, the crystallizer was filled with a 4 wt.% sodium sulphate aqueous solution. Thereafter, the temperature logger software was opened and set up prior to the starting the cooling unit. The cooling unit was set to the required temperatures and started in order to allow the coolant to flow. The temperature logger was started, followed by the starting of the agitator set at 230 rpm. Thermodynamic modelling using OLI Stream Analyser$^{TM}$ 10.0 [19] software predicted that salt would crystallize out of solution from $-1.03$ °C. Ice was predicted to crystallize out of solution at $-1.15$ °C; therefore, the eutectic temperature for a binary $Na_2SO_4$ brine is $-1.15$ °C as both ice and salt were crystallizing out of solution at this point. This meant the experiments needed to be conducted at temperatures below $-1.15$ °C. When T1 = $-1.15$ °C, 1 g of reagent grade $Na_2SO_4.10H_2O$ was added as seeds to control the point of salt crystallization and to ensure crystallization of ice was under eutectic conditions. It was found that supersaturations less than 0.2 °C allowed for large crystals [22] so, the seeding temperature for ice of $-1.3$ °C (supersaturation = 0.15 °C) was chosen to promote large product crystals. When the bulk temperature reached $-1.3$ °C, ice seeds were added to the system at 6 wt.% for the $\Delta T_{LMTD}$ experiments, and at the test level seed loading (0.1, 3, 6, 9, 12 wt.%).

The seeds were harvested from the ice shaving machine just in time for seeding in the batch crystallization experiments. Crystallization continued for 1 h and 15 min after seeding. Thereafter the coolant flow was stopped by switching the cooling unit off. This was followed by stopping the agitator.

The suspension was transferred from the crystallizer into a cooled 2 L beaker and then to a cooled Imhoff cone. The Imhoff cone and beaker were kept overnight in a cooled room at 2.5 °C and then in a freezer at $-18$ °C for 7 min before harvesting. This was done to minimize melting. The suspension was then stirred for 10 min to allow for the disengagement of salt from ice in a cooled room at 2.5 °C. Ice was harvested onto a Buchner funnel with 11 µm filter paper connected to a vacuum pump for filtration. A ratio of 0.5:1

of wash water: ice product was used during the experiments. The filtered ice and salt were weighed, and the masses were recorded.

The scale layer was melted off the wall of the crystallizer by raising the cooling unit set point to 20 °C. Thereafter, the melted ice scale was collected in a measuring cylinder and weighed. The mass of the scale layer was recorded.

10 wt.% of the ice product was collected as a sample from several positions in the beaker and transferred to a cooled, insulated flask for imaging. The procedure carried out for the seed size and shape characterization was followed for the ice product from all the experiments.

## 4. Results and Discussion

### 4.1. Effect of $\Delta T_{LMTD}$ on Ice Yield

The effect of $\Delta T_{LMTD}$ on ice yield is shown in Figure 5.

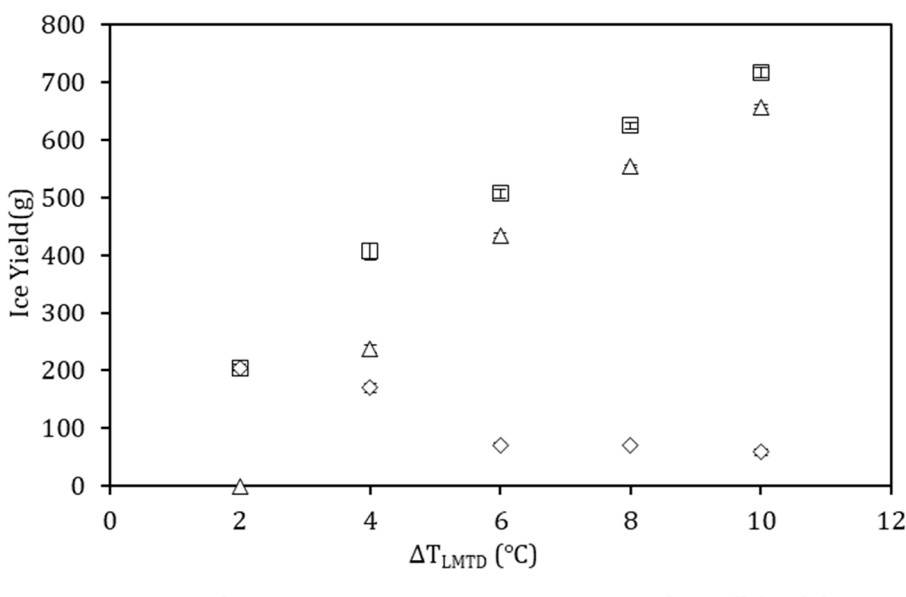

**Figure 5.** Effect of $\Delta T_{LMTD}$ on total ice yield.

The total yield of ice increased as the $\Delta T_{LMTD}$ increased. The total ice yield was the sum of the ice formed in suspension and ice formed on the wall (scale ice), with majority of the total ice yield being scale ice. The ice on the wall (scale layer) made up an increasing amount of the ice yield as $\Delta T_{LMTD}$ increased, with the scale contributing $\geq 91\%$ of the total yield at the highest $\Delta T_{LMTD}$ of 10 °C.

The observed increase in the yield can be attributed to an increase in the net heat transfer rate as the $\Delta T_{LMTD}$ increased i.e., the increase in the driving force due to the higher $\Delta T_{LMTD}$ outweighed the reduction due to scale formation. This is reflected in the values of Q (heat transfer rate) in Figure 6. The Q values were calculated using the measured coolant flow rate and temperatures, and Equation (6). The Q value presented here was the average taken from the point of seeding until the end of the experiment. From Figure 6, it can be seen that Q (the heat transfer rate) increased with increasing temperature driving force up to 6 °C, but then levelled off.

Vaessen and co-workers [13] found that, even though the increase in $\Delta T_{LMTD}$ increased the overall heat transfer rate, it also resulted in a decrease in the process heat transfer coefficient [13]. In their case, it was because the increase in $\Delta T_{LMTD}$ resulted in an increase in the solid content in the crystallizer (from approximately 6 vol.% solids to 10 vol.% at higher subcoolings).

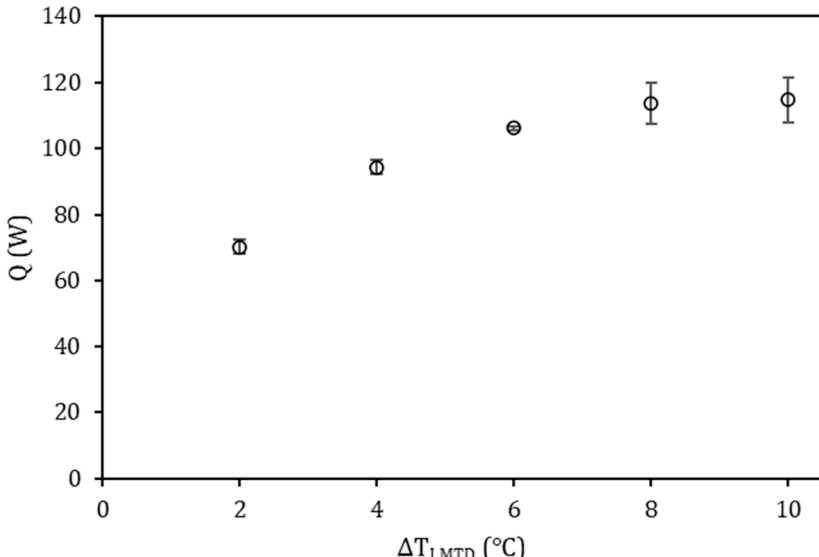

**Figure 6.** Effect of $\Delta T_{LMTD}$ on heat transfer rate (Q).

In this work, as $\Delta T_{LMTD}$ increased, although the total amount of ice formed did increase, the yield of ice in suspension decreased, while the yield of ice on the wall increased. This is illustrated in Figure 5. This was because, as the $\Delta T_{LMTD}$ increased, the cooled wall became colder and therefore the layer closest to the wall experienced the highest supersaturation. This led to the formation of a layer of scale ice on the wall.

The presence of the ice scale layer increased the thermal resistance, as demonstrated by the reduction in the overall heat transfer coefficient, U, illustrated in Figure 7. The instantaneous U values at each time interval were calculated using Equation (2), with the measured coolant and bulk temperatures, the heat transfer area and the Q values calculated earlier. Although Equation (2) applies to steady state conditions, the instantaneous measured values at each time interval can be used as an approximation.

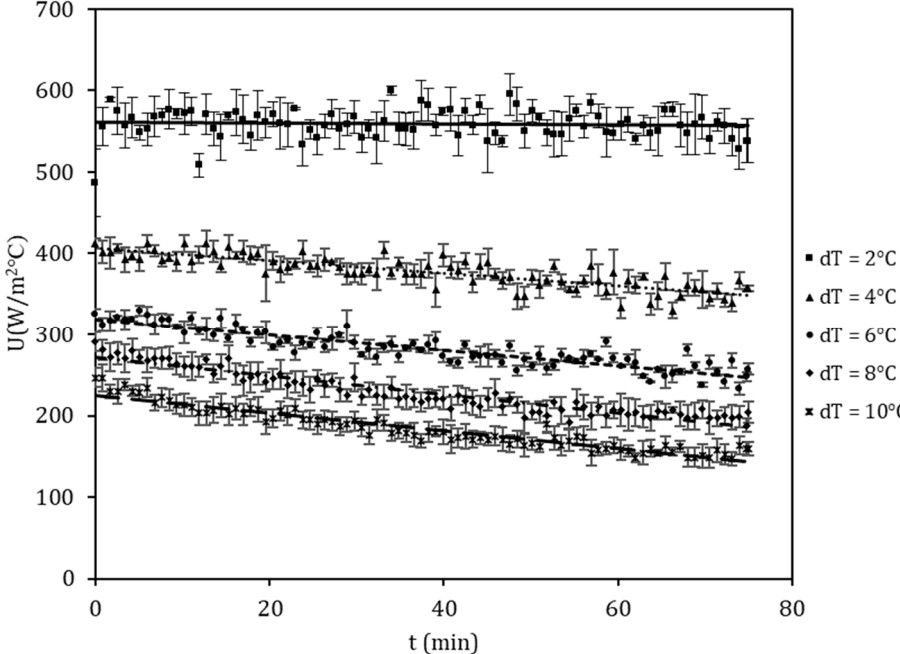

**Figure 7.** Effect of $\Delta T_{LMTD}$ on overall heat transfer coefficient (U).

Individual overall heat transfer coefficient (U) values were calculated for each set of temperature readings, which were 51 s apart as illustrated below in Figure 7, which shows the U values from the point of seeding (t = 0).

The reduction in the overall heat transfer coefficient resulted in a reduced yield of ice in suspension as shown in Figure 5. The reduction in the overall heat transfer coefficient with the increase in $\Delta T_{LMTD}$, is the reason for the plateau in the heat transfer rate that was observed in Figure 6. This reduction was due to the presence of the scale layer. The scale layer increases thermal resistance between the bulk and the coolant. This is depicted in Equation (5).

Figure 8 shows how the U value developed over time, for a single experiment for each and with changing $\Delta T_{LMTD}$. The other 2 repeats for each $\Delta T_{LMTD}$ follow the same trend but could not be combined (as in Figure 7) as the cooling time (time before seeding) varied for each experiment depending on the room temperature. Each point of seeding is indicated by a red data point for the specific $\Delta T_{LMTD}$.

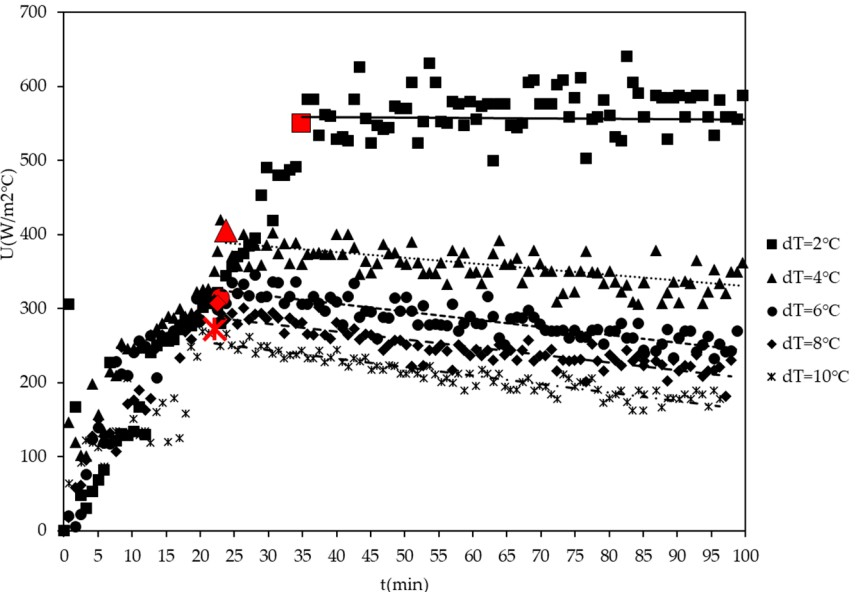

**Figure 8.** Effect of $\Delta T_{LMTD}$ on overall heat transfer coefficient (U), including pre-seeding time interval.

From Figure 8, it can be seen that a maximum U value was achieved at the point of seeding for $\Delta T_{LMTD}$ above 2 °C. The maximum attained U value decreased as $\Delta T_{LMTD}$ increased. This is most likely due to the rapid formation of the scale layer forming at the higher $\Delta T_{LMTD}$ values, which retarded the heat transfer rates, and prevented the U values from reaching the higher values.

This work indicates that, in EFC, although the use of a higher temperature driving force will potentially enhance the production rate, it will also enhance the tendency of the crystallizer to produce ice scale, thus substantially reducing efficiency. Therefore, an optimum temperature driving force for the system needs to be selected.

This work also shows how important it is to prevent scale formation, and the extent to which it significantly compromises the heat transfer rate, and thus the potential production rate of the crystallizer.

The following section explores the use of **seed loading** as a potential means of reducing ice scaling.

### 4.2. Effect of Ice Seed Loading on Ice Yield

The effect of ice seed loading on the yield of ice at a constant $\Delta T_{LMTD}$ of 4 °C is shown in Figures 9 and 10. As in the previous experiments, the ice yield was split between the ice in suspension and the ice on the wall (ice scale) see Table A1 in Appendix A.

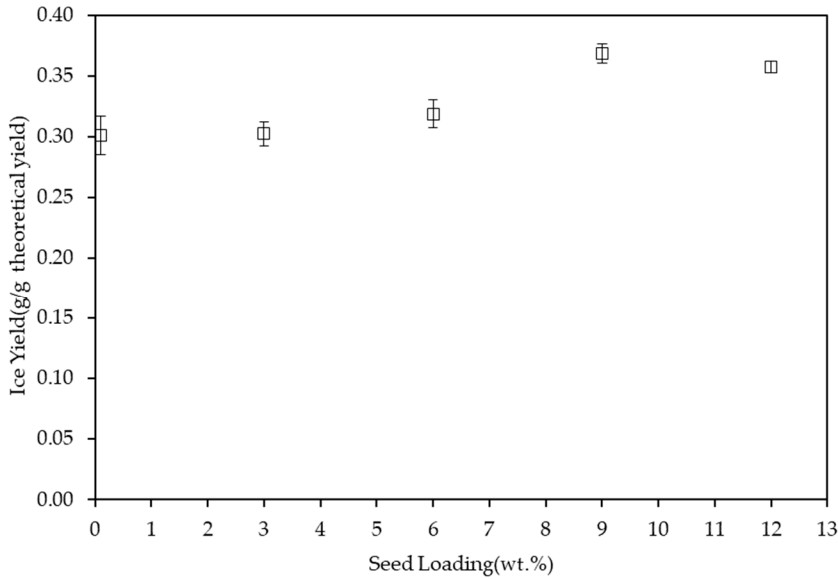

**Figure 9.** Effect of seed loading on total ice yield.

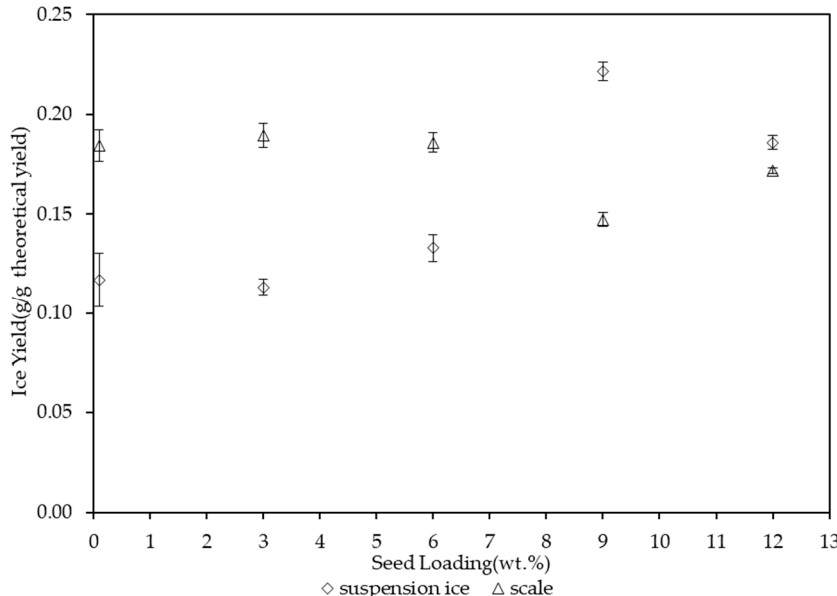

**Figure 10.** Effect of seed loading on suspension and scale ice yield.

The total yield of ice increased with increasing seed loading, with a maximum yield (37%) obtained at a seed loading of 9 wt.%. Figure 10 shows that, for all seed loadings, the amount of ice formed in suspension and the amount of ice formed as scale on the wall were of the same order of magnitude. This is likely because the $\Delta T_{LMTD}$ of 4 °C is relatively small, so the supersaturation at the wall was not very high. This meant that growth of both the scale and the suspension ice was gradual, unlike for higher $\Delta T_{LMTD}$ values.

For the lower seed loadings (0.1, 3 and 6 wt.%), more ice was formed on the wall, and for the higher seed loadings of 9 and 12 wt.%, more ice crystallized in suspension. At a seed loading of 9 wt.%, 50% more ice was formed in suspension than on the wall.

The amount of ice formed in suspension gradually increased as the seed loading increased, with the peak at a seed loading of 9 wt.%, whereafter the amount of ice in suspension decreased. The magma density increased as seed loading increased, which is to be expected, as the magma consists of the sum of the original seeds and the ice that crystallized during the experiment.

Figure 11 shows the final magma densities at each seed loading. It can be noted that, at the highest magma density of 31 wt.% (at seed loadings of 9 and 12 wt.%), ice production in suspension was maximized, while ice formation on the wall as scale was minimized.

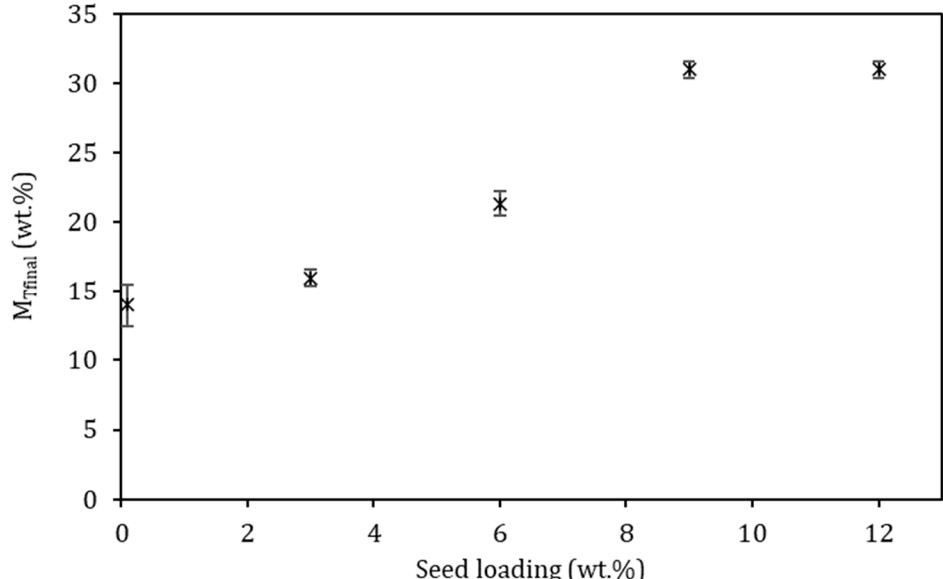

**Figure 11.** Effect of seed loading ($M_{T0}$) on final magma density ($M_{Tfinal}$).

The amount of ice formed on the wall remained relatively constant across the full range of seed loadings, with a minimum being measured at a seed loading of 9 wt.%. This suggests that adhesion-induced scaling, due to increased ice crystal-wall interaction, was not a significant phenomenon. The higher number densities of ice crystals in suspension at higher seed loadings would increase the chances of ice-wall collisions, and hence probability of ice adhesion onto the wall. Although a slight increase in the amount of the ice scale was observed when the seed loading was increased from 9 wt.% to 12 wt.%, the amount of scale was still smaller than the quantities recorded for other seed loadings (0.1, 3 and 6 wt.%), and hence was considered insufficient evidence of adhesion.

The secondary effects of the increase in seed loading on hydrodynamics (Re) and the process heat transfer coefficient ($\alpha_{process}$) were evaluated. The calculated Re values are illustrated in Figure 12, which shows the calculated Re at each seed loading upon introduction of the seeds, $M_{T0}$ and Re values computed at the final magma density, $M_{Tfinal}$.

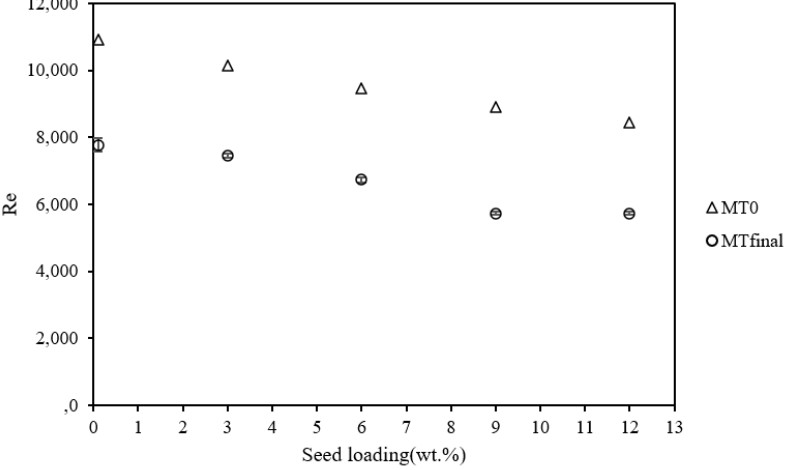

**Figure 12.** Effect of seed loading on Reynolds number.

Figure 13 shows the initial and final values of the process side heat transfer coefficient, $\alpha_{process}$. The decrease in $\alpha_{process}$ as the seed loading increased was considered insignificant.

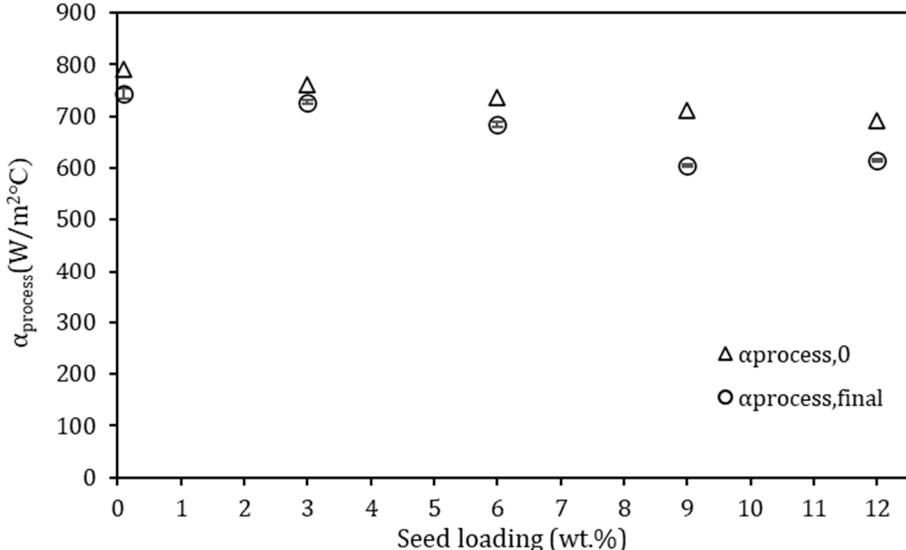

**Figure 13.** Effect of seed loading on process side heat transfer coefficient.

It was established that the change in seed loading did not have a significant effect on either of Re or the process side heat transfer coefficient. Therefore, the observed reduction in scaling as the seed loading increased was attributed to the increase in surface area of ice seeds in the crystallizer. It is proposed that higher surface area of the seeds enhanced the crystallization rate of ice in the bulk, and subsequently reduced rates of crystallization at the wall, hence the observed decrease in the amount of ice on the wall.

The presence of seeds reduces the energy barrier for nucleation and this effect becomes more pronounced as more seeds are provided [22]. Therefore, the increase in-seed loading possibly accelerated the rate of consumption of supersaturation in the bulk, thereby reducing the supersaturation available for heterogeneous nucleation and crystal growth on the wall. The decrease in the degree of supersaturation near the wall possibly suppressed the rates of heterogeneous nucleation and growth. In order to gain insight into the dominant mechanisms of ice crystallization in suspension, particle size analysis of the ice product harvested from the suspension was conducted.

Figure 14 shows the size distribution of the formed ice crystals at the seed loadings investigated in this study. Table 1 shows the d50 of the product ice crystals as the seed loading was increased.

Figure 15 shows the micrographs of the produced ice, which formed as large, flat discs. The salt crystals are visible in the micrographs as small white translucent particles.

From Figure 14 and Table 1 it is apparent that the particle size distributions of the suspension ice product at different seed loadings were unimodal. All seed loadings produced very similar ice product sizes, with the d50 and other size statistics of the ice product being very similar at all seed loadings. The change in seed loading hardly affected the final product size at all.

Analysis of the particle size results shows that growth occurred on the ice seeds [18]. However, the increase in the quantity of ice crystals (especially when a seed loading of 0.1 wt.% was used) suggests that nucleation and growth of more ice crystals (other than seeds) also occurred during the crystallization period. The similar particles sizes at the different seed loadings and the absence of peaks in the small size range seem to suggest that both the formed and seed crystals grew quite significantly during the crystallization period. Although it is evident that nucleation occurred during crystallization, relatively large ice crystals (>1 mm) were harvested at the end of the experiment. This suggests that either (a) nucleation occurred mainly at the beginning of the crystallization period,

allowing sufficient time for growth or size enlargement during the crystallization period or (b) that the nucleation rate was slower than the growth or size enlargement rate, i.e., the conditions favored growth or size enlargement over nucleation.

Further qualitative analysis of the crystallization mechanisms of ice in suspension was attempted by comparing the tested seed loadings to the critical seed loading value ($W_s^*$), recalculated from the maximum experimental yield and Equation (7). As shown in Figure 16, all the seed loadings were below the critical seed loading value. This supports the occurrence of both nucleation and growth as physically observed during experiments. However, all the distributions were unimodal which is inconsistent with the findings of Doki et al. [17], who observed bimodal product distributions when the seed loading was below the critical value. In this work, it is possible that, for all seed loadings, the full development of the particle size distribution of the suspended ice might not have been achieved since some of the ice crystallized onto the wall.

Since the particle size distributions of the suspension ice product were similar for all seed loadings, the higher yield of suspension ice as the seed loading increased was attributed to the increased number of ice crystals.

Overall, increasing the seed loading promoted crystallization of ice in the bulk and reduced scale formation on the wall. This was attributed to preferential consumption of supersaturation in the bulk as more surface area of seeds was available at higher seed loadings. The decrease in the amount of the scale on the wall is consistent with the observations by Leyland and collaborators [23].

The final size distribution of the suspension ice product was almost independent of the initial seed loading. Size analysis of the suspension ice product showed results that were expected when the seed loading is greater than the critical seed loading, where only growth of the seeds is favored. This result was unexpected, as both nucleation and growth of ice were evident. It was evident that, in some cases, the provided seeds induced nucleation and that growth/size enlargement of both seeds and nucleated ice crystals occurred in the suspension, resulting in unimodal final ice product size distributions. It is apparent that both nucleation and growth/size enlargement processes enhanced ice crystallization in the suspension, thereby retarding the rate of scale formation on the crystallizer wall. The individual contributions of these two processes to the reduction in scaling could not be quantified but it appeared as if growth/size enlargement, which is expected to be predominant at higher seed loading, was more effective at suppressing scaling.

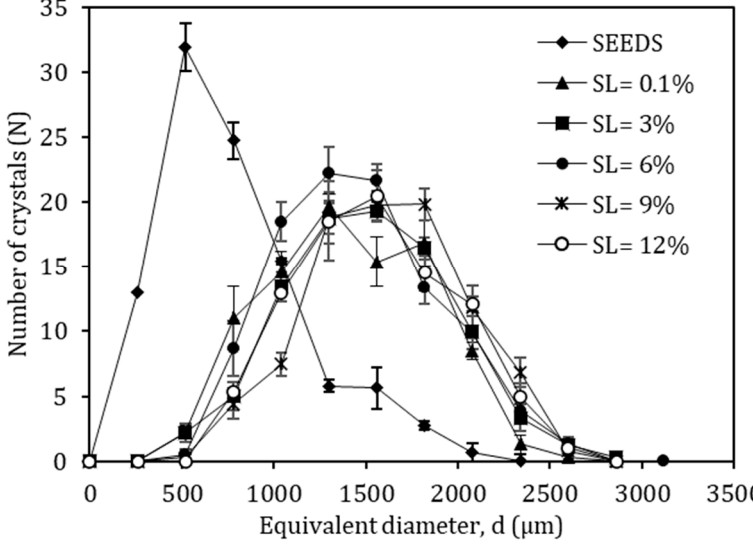

**Figure 14.** Effect of ice seed loading on ice product size.

**Table 1.** Effect of ice seed loading on ice product size.

| Seed Loading (wt.%) | Seeds | | | Ice Product Sizes | | |
|---|---|---|---|---|---|---|
| | $d_{10}$ (µm) | $d_{50}$ (µm) | $d_{90}$ (µm) | $d_{10}$ (µm) | $d_{50}$ (µm) | $d_{90}$ (µm) |
| 0.1 | | | | 1200 | 1786 | 2375 |
| 3 | | | | 1075 | 1636 | 2233 |
| 6 | 260 | 573 | 1261 | 1051 | 1560 | 2020 |
| 9 | | | | 1188 | 1744 | 2309 |
| 12 | | | | 1112 | 1663 | 2275 |

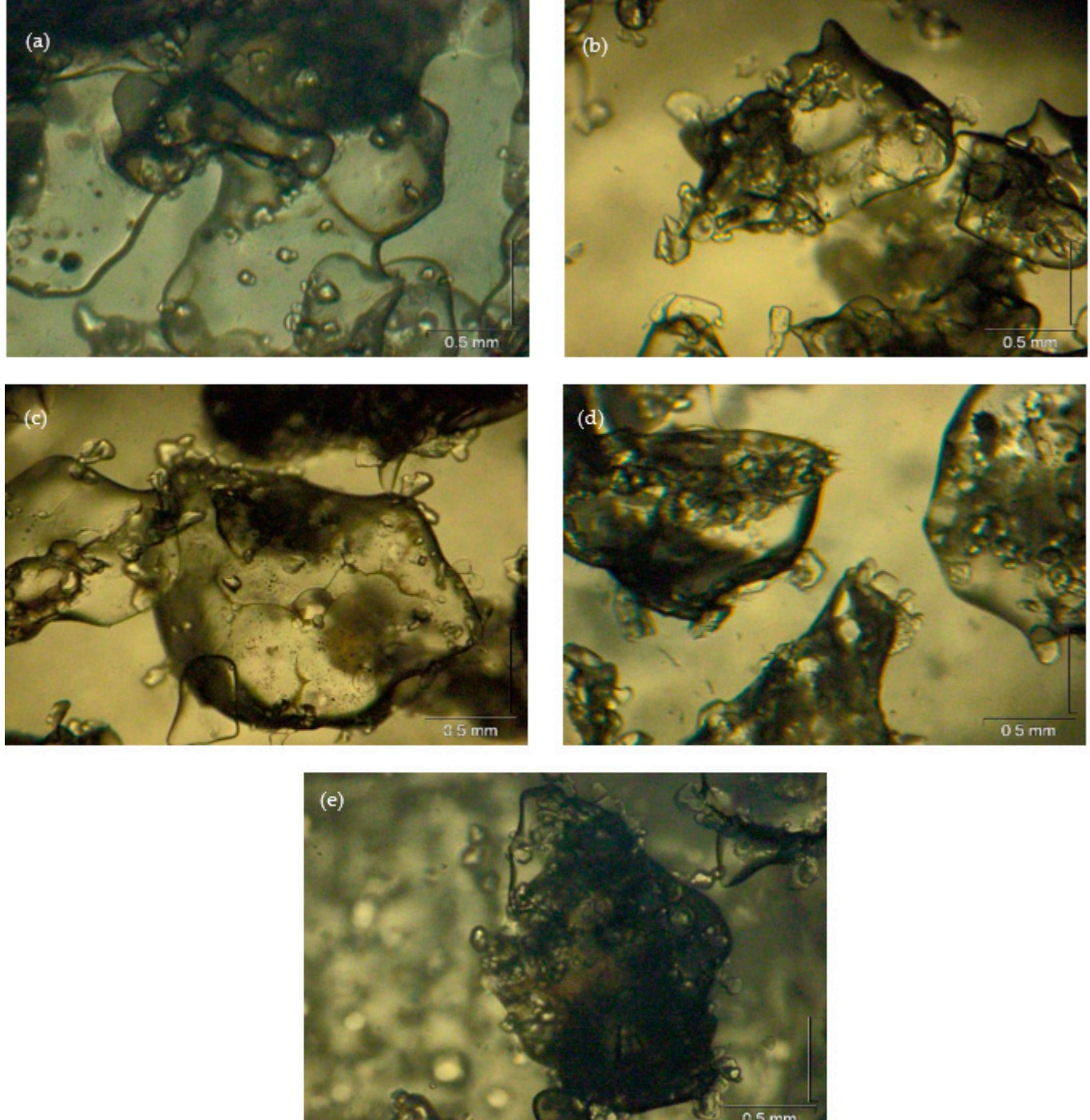

**Figure 15.** Typical ice product at SL = 0.1 (**a**), 3 (**b**), 6 (**c**), 9 (**d**) and 12 wt% (**e**).

Although increasing the seed loading in the range investigated in this study (0.1–12 wt.%) did not eliminate the scale layer, the observed reduction in the amount of scale at seed loadings greater than 6 wt.% showed that seeding can be a potential scale-management solution during EFC. The authors recommend further investigations, such as repeating the study at higher $\Delta T_{LMTD}$, or in continuous mode, to develop better understanding of the interaction between bulk ice crystallization and ice scale formation on the crystallizer wall in future studies as this could not be fully elucidated in this study. These further studies would better inform the use of ice seeds in continuous industrial studies. Knowledge on whether crystallization in the bulk and on the wall occur simultaneously (in parallel) or sequentially (in series) or both could be important in the design and operation of scale-free EFC systems.

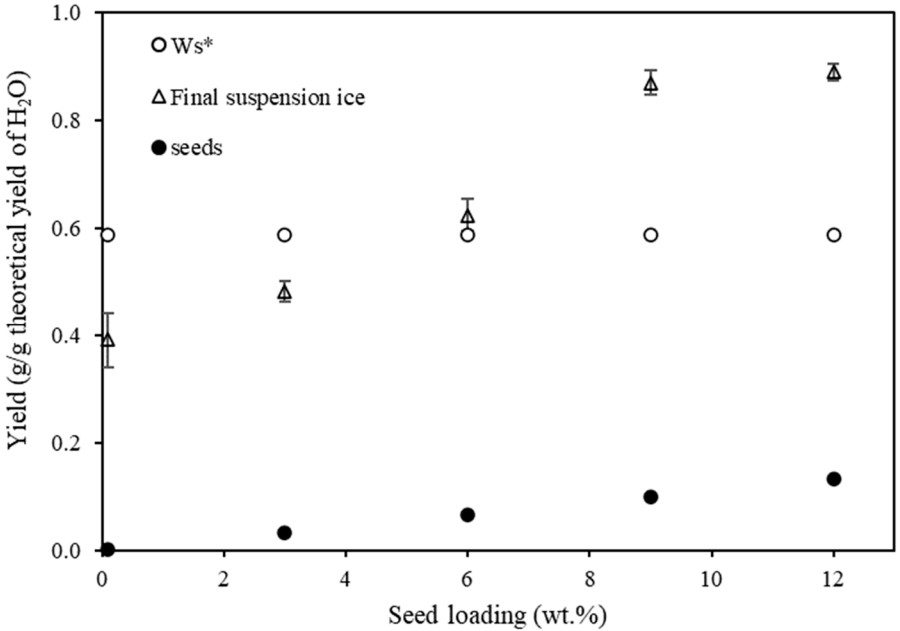

**Figure 16.** Effect of seed loading on final suspension ice and critical seed loading ($W_s$*).

Further investigations should also be conducted in scale-free eutectic systems to establish the potential of improving the particle size distributions of the ice product through seeding.

## 5. Conclusions

The aim of this study was to investigate and understand the effect of heat transfer driving force, $\Delta T_{LMTD}$, and ice seed loading (SL) on ice scaling from a dilute brine using Eutectic Freeze Crystallization.

The ice yield was split between ice harvested from suspension and ice formed on the wall. The yield of ice in suspension decreased with increasing $\Delta T_{LMTD}$, whilst the yield of ice on the wall increased. This was because, as the $\Delta T_{LMTD}$ increased, the increased supersaturation at the wall led to the formation of a scale layer. The presence of the scale layer reduced the overall heat transfer coefficient. The specific heat transfer rate, instead of increasing linearly, plateaued at higher $\Delta T_{LMTD}$ values, due to the additional thermal resistance from the ice formed on the wall.

- The ice on the wall (scale layer) made up an increasing amount of the ice yield as $\Delta T_{LMTD}$ increased, with the scale contributing $\geq 91\%$ of the total yield at the highest $\Delta T_{LMTD}$ of 10°C.
- The ice on the wall caused a significant reduction of up to 67% in the heat transfer rate between the coolant and the bulk, demonstrating the significance of avoiding scale formation in EFC.

It was found that, as seed loading increased, the total yield of ice increased, with a maximum ice yield obtained at a seed loading of 9 wt.%. For all seed loadings, the amount of ice formed in suspension and the amount of ice formed as scale on the wall were of the same order of magnitude.

- The amount of ice formed in suspension gradually increased as the seed loading increased while the amount of ice formed on the wall remained relatively constant across the full range of seed loadings. A maximum in the amount of ice in suspension and a minimum in the amount of ice on the wall were observed when a seed loading of 9 wt.% was used. At this seed loading, there was 50% more ice in the bulk than on the wall.
- The observed reduction in scaling as the seed loading increased was attributed to the provision of more surface area at higher seed loading, which enhanced ice crystallization in the bulk and suppressed heterogeneous nucleation at the wall.
- Although the yield of ice in suspension increased as seed loading increased, the final particle size distribution of the ice was found to be unimodal and independent of the initial seed loading. It was concluded that both nucleation and growth/size enlargement of ice crystals occurred during the crystallization period.

Finally, it can be concluded that an increase in the temperature driving force led to an increase in ice formed on the wall. However, ice seeds could be used to mitigate the formation of ice scale and promote ice crystallization in the bulk.

**Author Contributions:** Conceptualization, J.C. and A.E.L.; Data curation, A.S.; Funding acquisition, A.E.L.; Investigation, A.S.; Methodology, A.S.; Project administration, J.C. and A.E.L.; Resources, A.E.L.; Supervision, J.C. and A.E.L.; Writing—original draft, A.S.; Writing—review & editing, J.C. and A.E.L. All authors have read and agreed to the published version of the manuscript.

**Funding:** This research was funded by Coaltech Grant Number UCT 30271 and the Julian Baring Scholarship fund.

**Data Availability Statement:** Data is available in a publicly accessible repository at the University of Cape Town (https://open.uct.ac.za accessed on 11 August 2022) under Anotidaishe Spencer.

**Conflicts of Interest:** The authors declare no conflict of interest.

## Appendix A

**Table A1.** Seed loading ice yield results.

| SL (wt.%) | Suspension Ice | Scale | Total Ice |
|---|---|---|---|
| | | g/g Theoretical Yield | |
| 0.1 | 0.12 | 0.18 | 0.30 |
| 3 | 0.11 | 0.19 | 0.30 |
| 6 | 0.13 | 0.19 | 0.32 |
| 9 | 0.22 | 0.15 | 0.37 |
| 12 | 0.19 | 0.17 | 0.36 |

## Appendix B

Calculation of Re

Equation (A1) was used to calculate the Re for the suspension after seeding and at the final magma density. Where $\rho$ (kg/ m$^3$) was the density of the suspension, $v$ (m/s) was the velocity of the suspension, D (m) was the diameter of the impeller and $\mu$ (Pa.s) was the suspension viscosity.

$$Re = \frac{\rho_{suspension} N D_{impeller}^2}{\mu_{suspension}} \tag{A1}$$

where N ($s^{-1}$) is the rotational speed of the impeller, $\rho_{suspension}$ ($kg/m^3$) was the density of the suspension and $\mu_{suspension}$ (Pa. s) was the dynamic viscosity of the suspension.

Equation (A2) was used to calculate the viscosity of the suspension [24].

$$\mu_{suspension} = \mu_{brine}(1 + 2.5\phi) \tag{A2}$$

where $\phi$ was the solids volume fraction, $\mu_{brine}$ was the brine viscosity which was obtained from OLI Stream Analyser $^{TM}$ V10.0 software.

The density of the suspension was calculated based on the mass fractions, $x$, of the solids and the brine. This is shown in Equation (A3).

$$\rho_{suspension} = x_{solids}\rho_{solids} + x_{brine}\rho_{brine} \tag{A3}$$

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
