# Peer review of "Effect of Heat Transfer Driving Force and Ice Seed Loading on the Production of Ice and Salt from a Dilute Brine Treated Using Eutectic Freeze Crystallization"

_minerals, doi:10.3390/min12091094_

Round 1

Reviewer 1 Report

Comments for Draft- minerals 1843049

This paper explores the effect of Temperature driving force ΔTLMTD and ice loading on the production of ice formed on the container wall and the ice formed in the suspension brine. This work does good analysis of the ice size and ice images as well as the results discussions. Although this work emphasizes on EFC, the authors have ignored the analysis of salt crystallization and production within the process. Moreover, the study focused more onto the heat transfer and mass production with minor consideration on the produced ice quality and process efficiency in terms of salt concentration in ice, etc. There are many comments to bring good understanding and focusing the scope of this work. I would  accept this work under major major  correction. Here are some comments hoping to improve this work.

1. Experimental procedure part, the procedure is not very clear, a schematic figure for the experimental process is a better option.

2. Crystal size analysis is important part in this paper and the suitable setup and correct operation is primary to obtain the correct results.  However, the description of the size analyzer and the operation of the size analysis, such as the requirement of the samples, the condition of the measurement, are not clear enough. Please make this part (end of 3.2 experimental Procedure) clearer and move this part before Figure 3.

3. Figure 3 shows left figure the crystals are dispensed, middle figure shows the crystals are attached and the right figure shows the crystals are more and attached. Form the figures, it is difficult to say these three experiments are repeatable.

4. This work claims to focus on the Eutectic Freezing of dilute brine. However, from my knowledge, the Na2SO4’s saturation at 0 ℃ is around 4.6 wt.%. Then the 4 wt.% solution in this work is not dilute brine.

5. In part 3.1 Experimental Design, 2nd paragraph, the reference value for ΔTLMTD is not given, is it -1.15 or -1.3 ℃? Besides, Eutectic point should be one point or two points for this solution?

6. Adding reagent grade Na2SO4 10H2O may also affect the ice crystal growth and this work did not study the harvest salt. If no Na2SO4 10H2O is added, what will happen?

7.In 3.2 experimental Procedure, salt sample is collected for imaging and size analysis, but no corresponding results are discussed in the later section. Also, the ice scale is melted for sulphate analysis, the corresponding results are not shown. Salt pocket may form in the ice according to some reference, so the salinity analysis in the ice is important.

8. Figure 6, Figure 7 are same data but at different time periods, please combine the two figures and their  analysis. Please provide a mark,  such as an arrow,  to indicate the seeding moment in Figure 7.

9. A table to show the data for all experiments is needed for Figure 8 because you use 37% and 50% in the results, the experimental data cannot be read exactly from the figure.

10. Since the reduction of ice scaling on wall at ΔTLMTD=4 ℃, is not obvious, more experiments at higher ΔTLMTD is suggested to conducted because the scaling at higher ΔTLMTD is more serious and the reduction may be also obvious.

11. The use of supersaturation is rather confusing!!. This crystallization is under Eutectic point, the salt is already saturated in the solution and the solution cannot be further concentrated. The solution can be fully crystallized with time under Eutectic point. Please consider this and change the description.

12. Stressing on point 4, in line 143 in the original manuscript the authors claim eutectic freezing using 4wt.% of Na2SO4. However, they mention at later stage that they have performed dilution to the brine solution. No further information on the dilution step was mentioned. What is the initial brine concentration after dilution? What about the purity of the produced ice (or desalination efficiency)?

13. Some small mistakes:

1. Line 13 in same paragraph, Zeroà zero

2.Line 3 in paragraph above Eq 7, salt à ice

3. Slurry is better than magma.

4. Give numbering of the 3 figures in Figure 3 and Figure 13, such as a), b), c).

5. Wrong figure number used, “Figure 9” below Figure 8 should be “Figure 8”, After that, “Figure 10” should be “Figure 9”.

6. The Font of “Figure 10” above Figure 10 is not correct.

Author Response

The review points from the reviewers were addressed in red font underneath the point. Where the review note was addressed by a change to the manuscript, the line number where the change was made is referenced.

Reviewer 2 Report

The authors of this manuscript studied the effect of heat transfer driving force and ice seed load in eutectic freeze crystallization. This work could be of significant interest to the field, but I cannot recommend its publication in its current form, based on the following reasoning:

·     The introduction needs to be revised to a significant extent. The gaps in literature and the objectives of this work are not defined clearly. 

·    The novelty of this work is not apparent, which makes it difficult to justify whether this article contributes significantly to the existing knowledge on the field.

·      The authors do not justify the selection of the 4 wt.% Na2SO4 concentration. Is it common to work with this concentration in EFC? Is this practical? Should a wider concentration range be studied instead?

·   There is a contradiction between the background information presented in Section 2 and the ice loading values used by the authors in this work (lines 153-161). I would suggest expanding on the study from Kaboyashi and Shirai and help the reader understand why ice seed loadings between 0 and 12 wt.% makes sense. 

·      Lines 171-175: a batch reactor was used for simplicity, but the authors should explain and provide some insight as to how the results obtained in their setup could be useful in an industrial EFC setup / process. Also, what are the limitations of the setup used with regards to the results presented in this work.

·    Lines 198-204 and Figures 2 and 3: There is neither information regarding the equipment used to obtain these results nor any methodology used to obtain these results.

·      Lines 220-222: it is not clear what the authors mean to say in this part.

·   In Figure 4, the text of the y-axis is on top of the numerical values. I would suggest to add some space.

·      Based on Figure 7, the authors suggest a trend after the 28th minute (which is consistent with the data presented in Fig. 6). However, it is not discussed why this seems to happen at this particular time in almost all dT and what happens before. 

·      In Figure 9, the text of the x-axis title is on top of the numerical values. I would suggest to add some space.

·      The methodology and equipment used to obtain the images presented in Figure 13 were not presented in the manuscript. This is important information that should be added in the revised version.

·      Finally, the manuscript needs proofreading as there were a few grammatical and language errors.

Author Response

(The authors gave the same response as above.)

Round 2

Reviewer 1 Report

The authors satisfactorily responded to all the raised comments. 

Reviewer 2 Report

The authors have adequately addressed my comments. Therefore, I can recommend the acceptance of the revised manuscript.